# Line-Field Confocal Optical Coherence Tomography of Basal Cell Carcinoma: Systematic Correlation with Histopathology

**DOI:** 10.3390/diagnostics15233059

**Published:** 2025-11-30

**Authors:** Lucas Boussingault, Clément Lenoir, Alessandro Di Stefani, Simone Cappilli, Margot Fontaine, Gwendoline Diet, Makiko Miyamoto, Elisa Cinotti, Linda Tognetti, Javiera Pérez-Anker, Josep Malvehy, Susana Puig, Jean-Luc Perrot, Ketty Peris, Véronique del Marmol, Mariano Suppa

**Affiliations:** 1Department of Dermatology, Hôpital Erasme (HUB), Université Libre de Bruxelles, 1070 Brussels, Belgiumveronique.delmarmol@hubruxelles.be (V.d.M.);; 2Department of Dermatology, Institut Jules Bordet (HUB), Université Libre de Bruxelles, 1070 Brussels, Belgium; 3Dermatology Department, Hospital Clínic de Barcelona, Universitat de Barcelona, IDIBAPS, 08036 Barcelona, Spain; 4UOC di Dermatologia, Dipartimento di Scienze Mediche e Chirurgiche, Fondazione Policlinico Universitario A. Gemelli—IRCCS, 00168 Rome, Italysimo.cappilli@gmail.com (S.C.); ketty.peris@unicatt.it (K.P.); 5Departement of Dermatology, Catholic University of the Sacred Heart, 00168 Rome, Italy; 6Dermatology Unit, Department of Medical, Surgical and Neurological Sciences, University of Siena, 53100 Siena, Italy; 7Groupe d’Imagerie Cutanée Non Invasive (GICNI) of the Société Française de Dermatologie (SFD), 75008 Paris, France; 8Department of Dermatology, University Hospital of Saint-Étienne, 42055 Saint-Étienne, France

**Keywords:** basal cell carcinoma, histopathology, skin imaging, line-field confocal optical tomography

## Abstract

**Background/Objectives**: Basal cell carcinoma (BCC) is the most frequent skin cancer in Caucasian populations. While dermoscopy supports diagnosis, accurate subtype classification requires histopathology. Line-field confocal optical coherence tomography (LC-OCT) offers high resolution, adequate penetration, and three-dimensional imaging, bridging the gap between dermoscopy and histopathology. This study assessed the concordance between LC-OCT and histopathology for BCC criteria and subtypes. **Methods**: We retrospectively analyzed 127 histopathologically confirmed BCCs from the Departments of Dermatology and Pathology, Hôpital Erasme, Brussels. LC-OCT images and corresponding histopathological slides were evaluated. Objective analysis used a predefined checklist of LC-OCT criteria compared with histopathology. Subjective analysis consisted of independent side-by-side assessments of global resemblance by three observers with varying expertise. Concordance rates and κ statistics were calculated. **Results**: The objective analysis showed the highest concordance (≥80%) for lobules, blood vessels, bright cells, lobule location, and dermal-epidermal junction disruption. Intermediate concordance (50–80%) was found for hemispheric morphology, outer bright rim, stromal stretching, and parakeratosis. Inner dark rim and palisading showed low concordance (<50%). Subjective evaluations demonstrated strong resemblance between LC-OCT and histopathology (overall concordance 81.1%), ranging from 86.6% to 98.4% across observers. Interobserver agreement was slight overall (κ = 0.10, *p* = 0.02), with one moderate pairwise κ (0.41). **Conclusions**: LC-OCT demonstrates good concordance with histopathology for key diagnostic and subtype-discriminating BCC features. Despite variability in subtle criteria, the findings support LC-OCT as a clinically relevant tool for non-invasive diagnosis and management of BCC.

## 1. Introduction

Basal cell carcinoma (BCC) is the most frequent cutaneous cancer in Caucasian populations. Although metastases are exceptional, locoregional invasion and disfiguration are possible. Early diagnosis and subtype classification are, therefore, pivotal to reduce BCC morbidity and optimize treatment, which will indeed vary depending on the subtype of the BCC: while superficial BCC (sBCC) can be treated medically, other BCC subtypes should primarily be managed surgically [1]. Even though dermoscopy can orient the diagnosis of BCC [2], the precise determination of the subtype requires a histopathological examination that can only be obtained with a biopsy [3].

In vivo non-invasive cutaneous imaging has been developed to enhance dermatologists’ diagnostic performance while diminishing unnecessary invasive surgical procedures (either incisional or excisional) [4]. Until recently, the leading in vivo non-invasive cutaneous imaging techniques were represented by optical coherence tomography (OCT) and reflectance confocal microscopy (RCM) [5]. Both techniques present with advantages and disadvantages. Conventional OCT produces vertical skin images like histology and penetrates 1–2 mm, enough to visualize the epidermis and reticular dermis. However, its low resolution (7.5 µm) prevents cellular visualization [6]. High-definition OCT (HD-OCT) improves resolution (3 µm, cellular level) but with reduced penetration (0.57 mm) [7]. RCM offers superior lateral resolution (1 µm), allowing excellent cellular detail. However, it provides horizontal images—less comparable to histology—and has a limited penetration (0.25 mm), restricting visualization beyond the papillary dermis [8]. Line-field confocal optical coherence tomography (LC-OCT) was developed to combine high resolution with sufficient penetration depth [9,10] providing high resolution (1.1-µm axial and 1.3-µm lateral resolution), high penetration depth (0.5 mm) and three-dimensional (3D) imaging. The newer models of LC-OCT also combine an integrated dermoscopy in the probe, allowing in vivo dermoscopy-imaging correlations. Other techniques such as the high-frequency ultrasound, Raman spectroscopy or terahertz pulse imaging can be used for non-invasive skin diagnosis [11].

BCC does not appear exactly the same on OCT, RCM, and LC-OCT, although the underlying structures correspond to the same histopathologic features. On OCT [12] and HD-OCT [13], BCC typically appears as hyporeflective dermal or subepidermal lobules surrounded by a hyperreflective rim, corresponding to tumour nests and compressed collagen, respectively. Individual basaloid cells cannot be visualized due to the limited resolution. In contrast, RCM provides higher cellular resolution and reveals dark tumour islands with peripheral palisading and peritumoral clefts within a bright fibrous stroma [14]. Basaloid cells can be distinguished, but the horizontal (en face) view and limited penetration depth can hinder the evaluation of deeper components. Using LC-OCT, lobules of BCC are also described as being surrounded by an outer bright rim and a middle dark rim; however, the grey core of the lobule exhibits a distinctive “millefeuille pattern”, so named because it resembles the layered structure of the eponymous French pastry [15]. This pattern corresponds to a laminated arrangement in which the orientation of basaloid cells is parallel to the epidermis.

Suppa et al. described LC-OCT criteria for BCC and found independent predictors of the main BCC subtypes [superficial (sBCC), nodular (nBCC), and infiltrative/morpheaform (iBCC)] [15]. That study supported the hypothesis of a precise correlation between the cyto-architectural structures observed with LC-OCT and histopathology. Ruini et al., using 52 histopathologically confirmed BCCs, showed that the overall BCC subtype agreement between LC-OCT and conventional histology was 90.4% [16]. Recently, Mtimet et al. developed a diagnostic algorithm to discriminate across BCC subtypes and to differentiate them from imitators such as intradermal nevi and sebaceous hyperplasia [17]. This latest study also showed that LC-OCT increases the diagnostic accuracy of BCC by at least 12% compared with dermoscopy.

However, systematic comparisons between each LC-OCT criterion and its histopathological counterpart are lacking. We hypothesize a high level of concordance between LC-OCT and histopathology for BCC. The objective of this study was to demonstrate this hypothesis by objectively and subjectively comparing BCCs observed in LC-OCT and histopathology.

## 2. Materials and Methods

Study sample—LC-OCT (Deeplive, DAMAE Medical, Paris, France) images and histopathological slides of BCCs were retrospectively collected from the databases of the Departments of Dermatology and Pathology of Hôpital Erasme, HUB, ULB, Brussels (Belgium). Inclusion criteria were: clear histopathological diagnosis including subtype classification (sBCC, nBCC, iBCC, and mixed forms); availability of good-quality LC-OCT images/videos, as determined by author consensus; and availability of histopathological slides for digital photography. Approval was obtained from the local Ethical Committee, and all patients gave informed consent for their images to be anonymously used in this study.

LC-OCT examination—A single examiner acquired both vertical and horizontal LC-OCT images and videos as well as 3D reconstructions.

Histopathological slides—Haematoxylin–eosin–stained slides were imaged at ×4 and ×10 magnifications using an optical microscope connected to a computer equipped with OLYMPUS cellSens software, version 3.2 (The Hague, Netherlands).

Evaluation of the concordance—Objective and subjective evaluations were performed to assess the comparison between LC-OCT and histopathology.

For the objective evaluation, we used a checklist of criteria derived from previously published LC-OCT features and their histopathological correlates (Table A1) [11]. LC-OCT criteria were assessed on images, videos, and 3D reconstructions by three independent observers with varying levels of expertise in LC-OCT (Observer 1: lowest; Observer 2: intermediate; Observer 3: highest) and histopathology (Observer 1: lowest; Observer 3: intermediate; Observer 2: highest). All observers were blinded to clinical, dermoscopic, and histopathological data. Histopathological criteria were assessed on digitized slides by a single independent board-certified dermatopathologist, also blinded to clinical, dermoscopic, and LC-OCT information. Both LC-OCT and histopathological criteria were evaluated as dichotomous variables (absent/present), except for lobule location (connected to epidermis, separated, or both). Concordance rates between LC-OCT and histopathology were calculated for each criterion to quantify the agreement between imaging and histopathological assessment. LC-OCT evaluation included both individual assessments from each observer and a consensus evaluation, defined as agreement by at least two of the three observers.

For the subjective evaluation, representative histopathological and LC-OCT images were subsequently selected for each lesion by Observer 1 and displayed side by side on a PowerPoint presentation (Microsoft PowerPoint, version 2019, Redmond, WA, USA). Then, the three LC-OCT observers independently rated the overall resemblance between LC-OCT and histopathology as either weak or strong for each case. Interobserver agreement among the three LC-OCT readers was assessed using Fleiss’ κ statistics. Pairwise comparisons were further evaluated using quadratic weighted κ statistics, given the ordinal nature of the subjective categories [18].

Continuous variables were presented as medians with ranges. Categorical variables were presented as numbers with percentages. Different groups were compared using Pearson’s chi-square test. All statistical tests were two-sided, with *p*-values <0.05 considered statistically significant. Analyses were performed using STATA^®^ version 14 (StataCorp LP, College Station, TX, USA).

## 3. Results

### 3.1. Patient and Tumour Characteristics

A total of 127 histopathologically confirmed BCCs were included. Study lesions belonged to 90 patients [52 (57.8%) females and 38 (42.2%) males; median age 62 (34–87) years; no female/male differences (*p* = 0.36)]. Median number of BCC per patient was 1 (1–14). Of the 127 study lesions, 91 (71.7%) were of pure subtype [59 (46.5%) sBCC, 27 (21.3%) nBCC, 5 (3.9%) iBCC] and 36 (28.4%) of mixed subtype (Table 1). Significant associations were found between BCC subtype and body location: sBCC was more common on trunk/limbs while nBCC and iBCC on head/neck (*p* < 0.001); pure subtypes were more common on trunk/limbs while mixed subtypes on head/neck (*p* = 0.02).

### 3.2. Objective Evaluation

Consensus objective evaluations of the concordance between LC-OCT and histopathology are reported in Table 2.

The highest concordance rates (80–100%) between LC-OCT and histopathology were found for the following criteria: lobule (100%), blood vessels (100%), bright cells within epidermis (99.2%), lobule core (99.2%), lobule location (92.9%), bright cells within lobules (88.2%), disorganized epidermis (86.6%), round/ovoid morphology (81.9%), and disruption of the dermal-epidermal junction (DEJ) (81.1%).Intermediate concordance rates (50–80%) between LC-OCT and histopathology were found for: hemispheric morphology (73.2%), outer bright rim (72.4%), erosion/ulceration (71.4%), stromal stretching (65.4%), crust (64.6%), branched morphology (63.8%), parakeratosis (58.7%), stromal brightness (56.7%) and polymorphic morphology (55.9%).Low concordance rates (<50%) between LC-OCT and histopathology were found for: inner dark rim (48.0%) and palisading (7.9%). This last criterion was detected in 125/127 (98.4%) histopathological slides but in only 8/127 (6.3%) LC-OCT images/videos/3D reconstructions.

Individual objective evaluations are presented for each observer in Table 3. Compared to the other observers, palisading and stromal brightness were detected more by Observer 3; inner and outer rim less by Observer 2; and bright cells within lobules less by Observer 1.

### 3.3. Subjective Evaluation

Subjective evaluations (Figure 1, Figure 2 and Figure 3) showed a high resemblance between LC-OCT and histopathology, ranging from 86.6% (observer 1, the least experienced in histopathology) to 98.4% (observer 2, the most experienced in histopathology).Table 4 shows good concordance among the three observers in the subjective correlation assessment. Most cases were rated as ‘strong’ by all observers (86.6–98.4%), whereas the ‘weak’ category was rarely assigned. The overall LC-OCT/histopathology concordance rate was 81.1%. Interobserver agreement was slight but statistically significant (κ = 0.10, *p* = 0.02), reflecting variability in individual ratings.Table 5 summarizes the pairwise interobserver agreement of the subjective evaluations. A moderate agreement was observed between observer 1 and observer 3 (concordance rate 86.0%; κ = 0.41; *p* < 0.001). In contrast, only slight agreement was noted between observer 1 and observer 2 (concordance rate 80.1%; κ = 0.09; *p* = 0.09) and between observer 2 and observer 3 (concordance rate 85.4%; κ = 0.09; *p* = 0.13), despite relatively high concordance rates. This discrepancy between high concordance rates and low κ values likely reflects the well-recognized prevalence effect and differences in observer experience, both of which can substantially lower κ despite good absolute agreement.

On histopathology, the palisading (yellow circle) is easily distinguishable from the peritumoral clefting (yellow arrow). However, LC-OCT exhibits an inner dark rim (yellow arrow) that could either correlate to clefting, palisading, or both. This picture illustrates why palisading and the inner dark rim had the lowest concordance percentage between LC-OCT and histopathology.

## 4. Discussion

The existence of a visual correlation between non-invasive cutaneous imaging and dermatopathology represents a fascinating hypothesis to confirm, particularly for BCC due to the treatment diversity according to BCC subtypes. In this field, non-invasive imaging techniques such as RCM, conventional OCT, and HD-OCT have been previously proven to have some level of concordance with histopathology [13,19,20,21,22,23]. The intrinsic technical characteristics of the LC-OCT (i.e., high resolution and penetration; vertical and, more recently, 3D imaging) make it a very good candidate to investigate a concordance with histopathology in the field of BCC.

The sample included in this study seemed to reflect the common BCC distribution in Caucasian populations, as superficial BCCs were more frequently observed on the trunk and limbs, whereas nodular and infiltrative BCCs predominated on the head and neck [2,24,25]. This distribution is thought to reflect differences in sun exposure: intermittent exposure on the trunk and limbs favours the development of superficial BCCs, while chronic exposure on the head and neck promotes nodular or infiltrative subtypes.

The present study is the first to objectively evaluate the concordance between LC-OCT and histopathology for all known criteria of BCC. The highest concordance with histopathology was detected for dermal lobules featuring the characteristic millefeuille pattern, blood vessels and bright cells within the epidermis, which represent the most common LC-OCT criteria for BCC [15].

Lobules of BCC have been correlated with histopathology across various imaging modalities, (including RCM, conventional OCT, and HD-OCT) and with different terms such as ovoid structures, basaloid islands, or grey/dark structures. Their identification is central to BCC diagnosis, not only histologically but also through these non-invasive techniques. A key diagnostic criterion is the internal architecture of the lobule: the characteristic millefeuille pattern observed in LC-OCT correlates strongly with the dense basaloid cellularity seen in histopathology. This feature is essential for differentiating BCC from clinical mimickers such as sebaceous hyperplasia or low-pigmented dermal nevi [26,27]. Building on existing diagnostic algorithms developed with HD-OCT [7], our recent study introduced a new algorithm leveraging LC-OCT for the same purpose [17]. The identification of a lobule was emphasized as the primary feature to evaluate before proceeding with the algorithm. Furthermore, the study demonstrated that LC-OCT improved diagnostic accuracy by 24% compared to clinical examination and by 12% compared to dermoscopy. Additional research comparing LC-OCT with dermoscopy alone has consistently shown improved sensitivity and specificity, not only for detecting BCC and its subtypes, but also for reliably distinguishing them from their common clinical imitators [28,29].

Blood vessels seen on LC-OCT correlated extremely well with histopathology. In dermoscopy, they classically appear as arborizing telangiectasias and are a hallmark of the diagnosis of BCC [30]. They can also manifest as short fine telangiectasias, which are more commonly associated with the sBCC subtype [31]. It is possible to observe in vivo the red blood cells flowing through them, appearing as very small hyperreflective elements. The vascular architecture of these tumours provides excellent dermoscopy-imaging-histopathology correlations. It has also been shown that LC-OCT may offer practical clues for identifying the vascular nature of a lesion and supporting its differential diagnosis, even in 3D [32,33].

Bright cells within the epidermis or lobules correlated well with immunocompetent cells and activated melanocytes—as already shown in RCM [19]—but also with melanophages, which should therefore be considered in the interpretation of bright cells on LC-OCT. RCM could not distinguish immunocompetent cells from activated melanocytes when bright cells were observed, even when their morphology (large, small, or dendritic) was taken into account [19]. As LC-OCT has a lower resolution than RCM, we did not attempt to morphologically differentiate bright cells in this study.

The presence of a disorganized or pleomorphic epidermis above BCC lobules was a positive criterion for BCC in previous observations with HD-OCT [13]. However, epidermal pleomorphism was only rarely associated with BCC in this study, both on LC-OCT and histopathology: the better resolution of LC-OCT compared to HD-OCT is likely to account for this discrepancy. In any case, even for this criterion LC-OCT correlated well with histopathology in this study.

Importantly, the other well-correlated criteria were those useful for BCC subtype discrimination, i.e., lobule location, lobule morphology and stretching of the stroma. Indeed, it was previously shown that nBCC is independently predicted by round macrolobules separated from the epidermis and by stromal stretching; sBCC by hemispheric lobules connected to the epidermis; and iBCC by branched lobules [15]. Moreover, the disruption of the DEJ (due to the presence of hemispheric lobules appended to the epidermis) could be considered as another marker of sBCC and was again well-correlated with histopathology. Similarly, Ruini et al. [16] reported that nBCC were mainly characterized by tumour nests in the dermis and white hyper-reflective stroma but also by atypical keratinocytes, dark clefting, and prominent vascularization; sBCCs showed a thickening of the epidermis due to a series of tumour lobules with clear connection to the DEJ (string of pearls pattern); and iBCC was characterized by elongated hyporeflective tumour strands, surrounded by bright collagen (shoal of fish pattern). These results further highlight the potential of LC-OCT to direct the correct therapeutic decision for BCC (topical treatment versus surgical excision). Interestingly, among the different lobule morphologies, the round/ovoid showed the best LC-OCT/histopathology concordance, followed by hemispheric and then branched. This seems to suggest that iBCC represents—among the pure BCC subtypes—the most challenging one to diagnose, although Ruini et al. could identify the shoal of fish pattern associated with iBCC in 100% of their iBCC cases.

Among LC-OCT criteria that showed moderate concordance with histopathology were epidermal erosion/ulceration, crust, and parakeratosis (which do not seem particularly useful for BCC diagnosis) as well as stromal brightness: the histopathological correlation of this purely LC-OCT-based criterion is, however, misleading (i.e., only due to the concordance between negative observations) and should be interpreted bearing in mind that it has no histopathological counterpart. Other moderately correlated criteria were the lobule’s outer bright rim (possibly due to the peritumoral stroma compression exerted by the BCC lobule and previously described with HD-OCT) [13] and the stromal stretching.

Unexpectedly, two hallmark histopathological features of BCC—the inner dark rim (clefting) and peripheral palisading—exhibited the lowest concordance between LC-OCT and histopathology (Figure 4). The inner dark rim, previously identified as clefting in both RCM [34] and HD-OCT [35], and palisading, also described in RCM [14] and occasionally in HD-OCT [13,35] were not consistently recognized in LC-OCT images. One possible explanation is that, in some lesions, the histological palisading may have corresponded to the inner dark rim seen in LC-OCT, while peritumoral mucin deposition could be reflected by an additional, thinner and distinctly visible darker rim located between the inner and outer bright rims. Further studies are required to clarify the precise LC-OCT correlates of palisading and mucin deposition in BCC. Consistent with this uncertainty, palisading showed the highest interobserver variability in our study. However, these two criteria seem to have limited relevance for BCC diagnosis/classification under LC-OCT, which conversely is exclusively driven by the presence, shape and epidermal connection of lobules featuring the millefeuille pattern [17]. Indeed, clefting and palisading were not included in the diagnostic algorithm proposed by Mtimet et al. [17]. Newer LC-OCT prototypes provide improved image quality and may facilitate visualization of these features.

It is also worth mentioning that histological processing alters the morphology of BCC through fixation, dehydration, and embedding, leading to tissue shrinkage (~11–19%), distortion, and partial loss of hydration [36]. In contrast, LC-OCT provides in vivo imaging of hydrated tissue, displaying the natural architecture of those lesions. Regarding peritumoral mucin deposition, after histological processing, most mucin is washed away, resulting in clefts appearing as empty clear spaces separating the tumour from the surrounding stroma. Those clefts can be artifactually accentuated or collapsed.

Lastly, comparing our LC-OCT findings with those of Ruini et al. [16], several features appeared with similar frequency in both studies (e.g., lobules, ‘string of pearls’ and ‘shoal of fish’ patterns, blood vessels, and altered DEJ profile). However, marked differences emerged for others: peripheral palisading was rare in our series (6.3%) but frequent in Ruini’s (67.3%), and the prevalence of inner/outer rims, stromal brightness, and epidermal disorganization also varied substantially. Future integration of artificial intelligence, already promising for keratinocyte atypia [37] and DEJ undulation [10] in actinic keratosis may help standardize these assessments.

As for the subjective evaluations, the overall agreement was high as a strong resemblance between LC-OCT and histopathology was found in more than 4 BCCs out of 5. Interestingly, the two most experienced observers reported very high LC-OCT/histopathology concordance, whereas the least experienced observer tended to give slightly more pessimistic evaluations (despite being responsible for selecting the key histopathological images used for comparison with LC-OCT). However, the least experienced observer achieved very good concordance rates for the criteria most relevant to BCC diagnosis and subtyping—namely, the presence of lobules with a millefeuille pattern, lobule location, and lobule morphology. This finding is in line with recent evidence indicating that even brief targeted training can significantly improve diagnostic accuracy: in one study, a single hour of training increased the overall diagnostic rate of inexperienced readers from 48% to 76% [38]. The concordance rate for palisading, which is a subtle criterion, was higher for the most experienced observer in LC-OCT and corroborates the existence of a learning curve for this technique. Importantly, the relatively low κ values should be interpreted with caution, as κ is known to be influenced by prevalence and observer experience; the high concordance rates indicate that observers agreed in most cases despite these statistical limitations and should be interpreted as a strong sign of resemblance between LC-OCT and histopathology in the field of BCC.

From a clinical standpoint, BCC often occurs in areas where surgery is challenging, such as the nasal pyramid, and where margin assessment can be difficult [39]. LC-OCT appears especially useful for BCC and margin evaluation [40], owing to its excellent resolution, improved penetration depth, convenient vertical sectioning, integrated dermoscopy, and compatibility with artificial intelligence–based analysis [41]. Altogether, these features make LC-OCT a versatile tool in various clinical scenarios. In contrast, high-frequency ultrasound and conventional OCT provide more depth information, while RCM, with its superior resolution, appears better suited for the assessment of melanocytic lesions. Beyond margin evaluation, LC-OCT can help differentiate between BCC subtypes—low-risk forms (superficial and nodular) and high-risk variants (infiltrative)—and guide management. The absence of clefting on RCM and OCT has been associated with infiltrative subtypes [42], and similar features can be observed with LC-OCT. Furthermore, in sBCC treated with non-surgical modalities such as topical imiquimod, LC-OCT can be used to monitor treatment response and confirm complete lesion disappearance [43]. Although no studies have yet specifically demonstrated that LC-OCT can reduce the number of biopsies required for the diagnosis of BCC, it is reasonable to anticipate that future studies may confirm such a thing, given that conventional OCT has already been shown to decrease the need for diagnostic biopsies in BCC [44,45]. Moreover, a recent prospective real-life study showed that RCM can reduce the number of biopsies for skin lesions in general, including BCC [46]. However, LC-OCT is not yet a validated tool for predicting patient outcomes such as recurrence; its current role remains primarily diagnostic and morphological rather than prognostic. Unfortunately, LC-OCT remains an expensive device that may not be accessible to all dermatology centres, and its economic impact has yet to be fully evaluated. However, the ongoing ECOBASO study in France is assessing the efficiency and economic implications of LC-OCT for the diagnosis and management of BCC [47]. Moreover, a real-world cost–benefit analysis of RCM for melanoma has shown that the cost per melanoma excised with standard care was nearly twice as high as with adjunctive RCM [48].

This study had several limitations, including its retrospective design, the lack of stratified analyses by BCC subtype, and the possibility that LC-OCT acquisitions were not performed at the exact same location or orientation as the corresponding histopathological sections. These spatial mismatches may have lowered the concordance between LC-OCT and histopathology. Moreover, the variable interobserver agreement underscores the need for larger, multicenter studies with standardized training.

## 5. Conclusions

In conclusion, our study explored the correlation between LC-OCT and histopathology for BCC. According to our results, a good concordance between the two techniques was detected, which supports our hypothesis underlying the study. Importantly, LC-OCT criteria that best correlate with histopathology were those important for BCC diagnosis and subtype discrimination. These data confirm the clinical relevance of LC-OCT for the diagnosis and management of BCC.

## Figures and Tables

**Figure 1 diagnostics-15-03059-f001:**
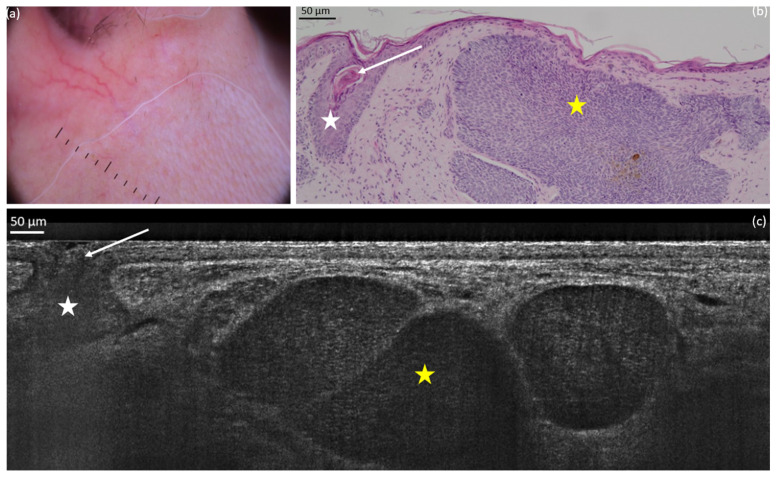
Nodular basal cell carcinoma (nBCC) on the upper right lip of a 50-year-old woman: dermoscopic (**a**) and histopathological (**b**) presentations; vertical line-field confocal optical coherence tomography (LC-OCT) image (**c**). The lobule of the nBCC (yellow star) is round/ovoid and separated from the epidermis. We can observe a hair follicle (white star): it can be mistaken as a BCC but it does not show the typical *millefeuille* pattern and the stroma involvement seen in BCC. The hair (white arrow) can be seen more easily on videos.

**Figure 2 diagnostics-15-03059-f002:**
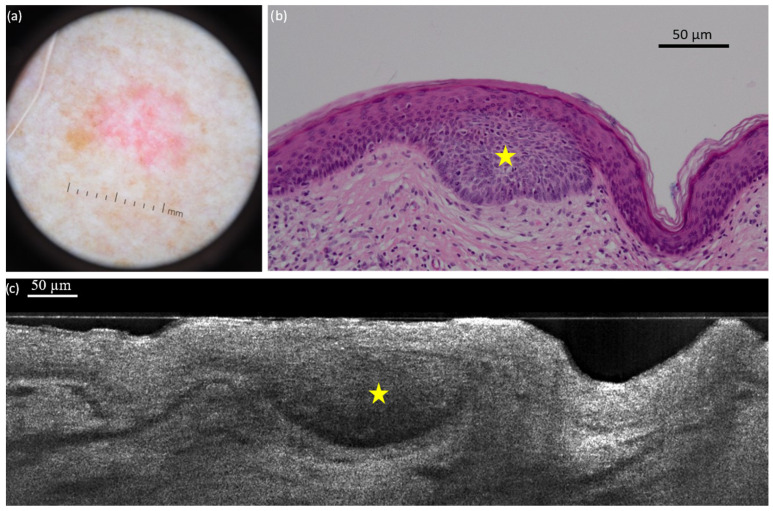
Superficial basal cell carcinoma (sBCC) on the right posterior shoulder of a 73-year-old man: dermoscopic (**a**) and histopathological (**b**) presentations; vertical line-field confocal optical coherence tomography (LC-OCT) image (**c**). The lobule (yellow star) is hemispherical and connected to the epidermis. Both modalities show the concavity of the skin next to the tumour.

**Figure 3 diagnostics-15-03059-f003:**
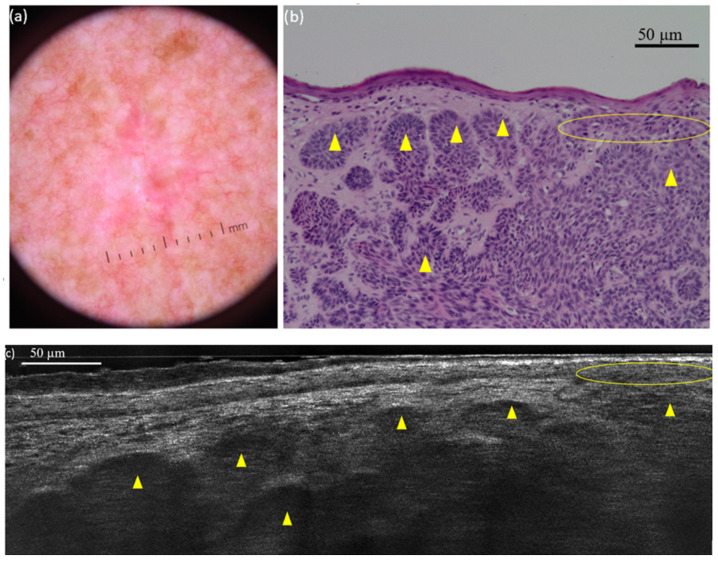
Nodular basal cell carcinoma (nBCC) on the right scapular region of a 66-year-old man: dermoscopic (**a**) and histopathological (**b**) presentations; vertical line-field confocal optical coherence tomography (LC-OCT) image (**c**). The tumour is composed of multiple round/ovoid-shaped lobules (yellow triangles). It is both separated and connected to the epidermis (yellow circle).

**Figure 4 diagnostics-15-03059-f004:**
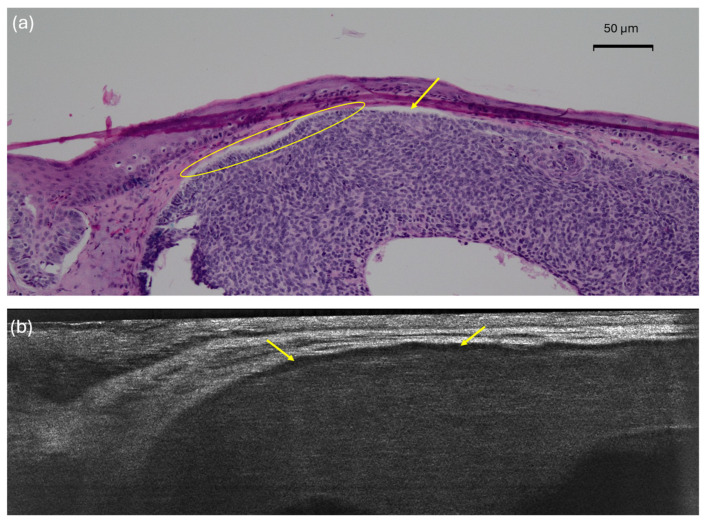
Nodular basal cell carcinoma (nBCC) on the upper right lip of a 50-year-old woman: histopathological (**a**) presentations showing clear palisading (yellow circle) and clefting (yellow arrows); vertical line-field confocal optical coherence tomography (LC-OCT) image (**b**) with an inner dark rim (yellow arrows) correlating with both the palisading and the clefting seen in histopathology.

**Table 1 diagnostics-15-03059-t001:** Number and frequency of subtypes according to body location.

	Overall (*n* = 127)	Head/Neck (*n* = 62)	Trunk (*n* = 37)	Limbs (*n* = 28)
Pure BCC subtypes	91 (71.7)	38 (61.3)	28 (75.7)	25 (89.3)
Superficial	59 (46.5)	11 (17.7)	24 (64.9)	24 (85.7)
Nodular	27 (21.3)	22 (35.5)	4 (10.8)	1 (3.6)
Infiltrative	5 (3.9)	5 (8.1)	0 (0)	0 (0)
Mixed BCC subtypes	36 (28.3)	24 (38.7)	9 (24.3)	3 (10.7)
Superficial and nodular	24 (18.9)	14 (22.6)	7 (18.9)	3 (10.7)
Nodular and infiltrative	7 (5.5)	5 (8.1)	2 (5.4)	0 (0)
Superficial and infiltrative	3 (2.4)	3 (4.8)	0 (0)	0 (0)
Superficial, nodular, and infiltrative	2 (1.6)	2 (3.2)	0 (0)	0 (0)

N (%) displayed in each box; BCC, basal cell carcinoma.

**Table 2 diagnostics-15-03059-t002:** Frequency of criteria and concordance rates between histopathology (independent evaluator) and LC-OCT (consensus among three evaluators).

	Histopathology	LC-OCT	Concordance (%)
Lobule	127 (100)	127 (100)	100
Lobule composition			
Core ^a^	127 (100)	126 (99.2)	99.2
Palisading	125 (98.4)	8 (6.3)	7.9
Inner rim ^b^	94 (74.0)	62 (48.8)	48.0
Outer rim ^c^	123 (96.9)	90 (70.9)	72.4
Lobule location			92.9
Separated from epidermis	16 (12.6)	18 (14.2)	
Connected to epidermis	45 (35.4)	41 (32.3)	
Both	66 (52.0)	68 (53.5)	
Lobule morphology			
Round/ovoid	107 (84.3)	92 (72.4)	81.9
Hemispheric	81 (63.8)	69 (54.3)	73.2
Branched	50 (39.4)	20 (15.8)	63.8
Polymorphic	102 (80.3)	56 (44.1)	55.9
Blood vessels	127 (100)	127 (100)	100
Stroma involvement			
Stretching	53 (41.7)	67 (52.8)	65.4
Brightness ^d^	0 (100)	55 (43.3)	56.7
Epidermal changes			
Parakeratosis	55 (43.3)	14 (15.2)	58.7
Disorganized epidermis	11 (8.7)	8 (6.3)	86.6
Disrupted DEJ	105 (82.7)	103 (81.1)	81.1
Bright cells ^e^			
within epidermis	127 (100)	126 (99.2)	99.2
within lobules	127 (100)	112 (88.2)	88.2
Other			
Erosion/ulceration	40 (31.5)	8 (6.3)	71.7
Crust	45 (35.4)	14 (11.0)	64.6

N (%) displayed in each box, unless otherwise stated. DEJ, dermal-epidermal junction. ^a^ Lobule’s core corresponds to dense cellularity (histopathology) and millefeuille pattern (LC-OCT). ^b^ Lobule’s inner rim corresponds to dark rim (LC-OCT) and clefting (histopathology) ^c^ Lobule’s outer rim corresponds to bright rim (LC-OCT) and perilobular stroma compression (histopathology) ^d^ Brightness cannot be visualized in histopathology. ^e^ Bright cells (LC-OCT) correspond to immunologically competent skin cells and activated melanocytes (histopathology).

**Table 3 diagnostics-15-03059-t003:** Frequency of criteria and concordance rates between histopathology and LC-OCT (for the three observers).

	Histopathology	Observer 1		Observer 2		Observer 3	
		LC-OCT	Concordance (%)	LC-OCT	Concordance (%)	LC-OCT	Concordance (%)
Lobule	127 (100)	127 (100)	100	127 (100)	100	127 (100)	100
Lobule composition							
Core ^a^	127 (100)	125 (98.4)	98.4	120 (94.5)	94.5	127 (100)	100
Palisading	125 (98.4)	3 (2.4)	3.9	9 (7.1)	8.7	32 (25.2)	26.8
Inner rim ^b^	94 (74.0)	60 (47.2)	52.8	20 (15.8)	29.1	119 (93.7)	69.3
Outer rim ^c^	123 (96.9)	111 (87.4)	85.8	53 (41.7)	43.3	88 (69.3)	70.9
Lobule location							
Separated from epidermis	16 (12.6)	19 (15.0)		18 (14.2)		22 (17.3)	
Connected to epidermis	45 (35.4)	34 (26.8)		40 (31.5)		45 (35.4)	
Both	66 (52.0)	74 (58.3)	89.8	69 (54.3)	93.1	60 (47.2)	91.7
Lobule morphology							
Round/ovoid	107 (84.3)	104 (81.9)	81.9	87 (68.5)	79.5	90 (70.9)	81.9
Hemispheric	81 (63.8)	77 (60.6)	71.7	33 (26)	59.1	79 (62.2)	73.2
Branched	50 (39.4)	41 (32.3)	63	14 (11.0)	62.2	19 (15.0)	63.0
Polymorphic	102 (80.3)	89 (70.1)	66.1	6 (4.7)	22.8	61 (48.0)	59.8
Blood vessels	127 (100)	127 (100)	100	127 (100)	100	127 (100)	100
Stroma involvement							
Stretching	53 (41.7)	59 (46.46)	62.2	84 (66.1)	56.7	64 (50.4)	64.6
Brightness ^d^	0 (100)	35 (27.56)	72.4	53 (41.7)	58.3	87 (68.5)	31.5
Epidermal changes							
Parakeratosis	55 (43.3)	27 (21.3)	59.1	4 (3.2)	56.7	29 (22.8)	54.3
Disorganized epidermis	11 (8.7)	20 (15.8)	78.7	2 (1.6)	89.8	14 (11.0)	83.5
Disrupted DEJ	105 (82.7)	81 (63.8)	68.5	102 (80.3)	81.9	93 (73.2)	74.8
Bright cells ^e^							
within epidermis	127 (100)	123 (96.9)	96.9	123 (96.8)	96.9	103 (81.1)	81.1
within lobules	127 (100)	84 (100)	66.9	121 (95.3)	94.5	109 (85.8)	85.8
Other							
Erosion/ulceration	40 (31.5)	17 (13.4)	69.3	3 (2.4)	69.3	15 (11.8)	67.7
Crust	45 (35.4)	39 (30.7)	65.4	0 (0)	64.6	24 (18.9)	64.6

N (%) displayed in each box, unless otherwise stated. DEJ, dermal-epidermal junction. ^a^ Lobule’s core corresponds to dense cellularity (histopathology) and *millefeuille* pattern (LC-OCT). ^b^ Lobule’s inner rim corresponds to dark rim (LC-OCT) and clefting (histopathology) ^c^ Lobule’s outer rim corresponds to bright rim (LC-OCT) and perilobular stroma compression (histopathology) ^d^ Brightness cannot be visualized in histopathology. ^e^ Bright cells (LC-OCT) correspond to immunologically competent skin cells and activated melanocytes (histopathology).

**Table 4 diagnostics-15-03059-t004:** Correlations between the three observers for each subjective score.

	Observer 1	Observer 2	Observer 3	*κ*	*p*-Value
**Subjective correlation**					
Weak	17 (13.4)	2 (1.6)	10 (7.9)		
Strong	110 (86.6)	125 (98.4)	117 (92.1)		
All evaluations	127 (100)	127 (100)	127 (100)	0.10	0.02

Overall agreement of 81.1%. N (%) displayed in each box, unless otherwise stated.

**Table 5 diagnostics-15-03059-t005:** Overall subjective correlations between two observers.

	Observer 1			Observer 2			Observer 3		
	Agreement (%)	*κ*	*p*-Value	Agreement (%)	*κ*	*p*-Value	Agreement (%)	*κ*	*p*-Value
Observer 1				80.1	0.09	0.09	86.0	0.41	<0.001
Observer 2	80.1	0.09	0.09				85.4	0.09	0.13
Observer 3	86.0	0.41	<0.001	85.4	0.09	0.13			

## Data Availability

The data that support the findings of this study are available from the corresponding author upon reasonable request.

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
