# Peer review of "Line-Field Confocal Optical Coherence Tomography of Basal Cell Carcinoma: Systematic Correlation with Histopathology"

_diagnostics, 2025, doi:10.3390/diagnostics15233059_

Round 1
Reviewer 1 Report
Comments and Suggestions for Authors
This study demonstrates that line-field confocal optical coherence tomography has substantial potential as a non-invasive imaging tool for the diagnosis and subtype classification of basal cell carcinoma, showing strong resemblance with histopathological findings in both objective and subjective assessments.
The introduction part of the manuscript can be improved and extended. Focus on OCT by expanding the BCC imaging aspects of this technique, along with comparison with other available, innovative techniques - https://doi.org/10.3892/etm.2021.10982.
In the discussion part, focus on the tumor margins and the evaluation with OCT, versus other imaging techniques, in vivo - Vatamanesku I, Parasca SV, Parasca OM, Vaida FA, Mehedinţi MC, Grosu F, Ciurea ME. Basal cell carcinoma of the nasal pyramid excision margins: a retrospective study. Rom J Morphol Embryol. 2019;60(4):1261-1268.
Does histological processing affect the tumor characteristics and are there any changes from the OCT imaging? Discuss.
What other patient prognostic factors could be evaluated with OCT? Discuss and suggest possible aggressive traits - https://doi.org/10.2147/CCID.S385213.
Can OCT predict patient outcome? Can it influence the treatment choice? Discuss.
Limitation of study - suggest that limited interobserver agreement highlights the need for larger, multicenter studies with standardized training - which is also a possible future research direction.
Comments on the Quality of English LanguageOverall English language editing.
Author Response
We thank the reviewer for the thorough and constructive evaluation of our manuscript. We address each comment in detail below and have revised the manuscript accordingly.
Reviewer comment:
The introduction part of the manuscript can be improved and extended. Focus on OCT by expanding the BCC imaging aspects of this technique, along with comparison with other available, innovative techniques - https://doi.org/10.3892/etm.2021.10982.
Response:
We thank the reviewer for this relevant suggestion. We have expanded the Introduction to better describe OCT principles and its application to BCC imaging, and added a comparison with other innovative non-invasive imaging modalities. We have incorporated the recommended reference. These additions can be found in the revised manuscript on page 02, lines 72-89.
Reviewer comment:
In the discussion part, focus on the tumor margins and the evaluation with OCT, versus other imaging techniques, in vivo - Vatamanesku et al., 2019.
Response:
We appreciate this valuable comment. The Discussion has been expanded to address the role of OCT in evaluating tumor margins and to compare its performance with other in vivo imaging techniques. We have integrated the reference suggested by the reviewer. These modifications are located on page 14, lines 387-397.
Reviewer comment:
Does histological processing affect the tumor characteristics and are there any changes from the OCT imaging? Discuss.
Response:
Thank you for pointing out this important aspect. We now explicitly discuss how histological processing (fixation, dehydration, embedding) may alter tumor morphology and potentially account for differences observed between OCT and histopathology. This discussion has been added on page 13, lines 356-362.
Reviewer comment:
What other patient prognostic factors could be evaluated with OCT? Discuss and suggest possible aggressive traits - https://doi.org/10.2147/CCID.S385213.
Response:
We thank the reviewer for this suggestion. We have added a section discussing potential prognostic features, referencing the proposed article. These additions appear on page 14, lines 397-402.
Reviewer comment:
Can OCT predict patient outcome? Can it influence the treatment choice? Discuss.
Response:
We agree that this is an important point. We have added a paragraph discussing the potential for OCT to guide management decisions, including predicting treatment response to non-invasive therapies. This section appears on page 08, lines 408-410.
Reviewer comment:
Limitation of study - suggest that limited interobserver agreement highlights the need for larger, multicenter studies with standardized training.
Response:
We appreciate this helpful suggestion. We have revised the Limitations section to emphasize that the limited interobserver agreement underscores the need for larger, multicenter studies and standardized training programs. This has been added on page 14, lines 421-422.
Reviewer 2 Report
Comments and Suggestions for Authors
Dear authors,
This is a very interesting paper, focusing on the correlation between line-field confocal optical coherence tomography (LC-OCT) and histopathology in a series of basal cell carcinomas of the skin.
This is an original paper, with a good comparison between a clinical and and histological technique, who leads to high concordance of some features and highlighting even some limitations of LC-OCT.
The abstract is clear.
The description of the background in the introduction is well-defined.
Data are presented clearly, with rigorous methodical approach.
Conclusions are consistent with the evidence and arguments presented and clinically relevant.
No ethical problems were found.
No inflammatory material is found
Figures and tables are valid and clear
References are appropriate for the type of the study herein reported.
Suggestions:
I suggest the authors to try adding some visual examples of the low concordance criteria
I suggest the authors to briefly described if the LC-OCT potentially reduce the number of cutaneous biopsies
Author Response
We sincerely thank the reviewer for the very positive assessment of our manuscript and for the constructive suggestions provided. We respond to each point below.
Reviewer comment:
I suggest the authors to try adding some visual examples of the low concordance criteria.
Response:
We thank the reviewer for this helpful suggestion. To improve clarity and better illustrate the features associated with lower concordance between LC-OCT and histopathology, we have added a forth figure illustrating how palisading and clefting can be difficult to assess on LC-OCT. This addition can be found on page 10.
Reviewer comment:
I suggest the authors to briefly describe if the LC-OCT potentially reduce the number of cutaneous biopsies.
Response:
We appreciate this valuable comment. We have added a short paragraph in the Discussion addressing the potential role of LC-OCT in reducing the number of skin biopsies. This addition can be found on page 14, lines 402-408.
Reviewer 3 Report
Comments and Suggestions for Authors
It would be good to mention the cost and training associated with LC-OCT imaging in some point in the paper. This comment does not have to specify exact price, but put into context the significant cost of the machine.
Author Response
We thank the reviewer for the constructive comment. Please find our detailed response below.
Reviewer comment:
It would be good to mention the cost and training associated with LC-OCT imaging in some point in the paper. This comment does not have to specify exact price, but put into context the significant cost of the machine.
Response:
We agree with the reviewer’s observation. We have added a paragraph in the Discussion acknowledging the substantial cost of LC-OCT devices. This content has been added on page 14, lines 410-416.
As for the training, there is a learning curve but, as demonstrated by Cinotti et al., 1 hour of training is enough to improve the diagnosis of BCC with LC-OCT. This content was discussed on page 13 line 376 - page 14 line 382.